# OpenReview forum: "The Latent Cause Blind Spot: an Empirical Study of Update Types and Their Collateral Effects on LLMs"
_ICLR.cc/2026/Conference — Submitted to ICLR 2026_

### Official Review · Reviewer_ECWU · 2025-10-30

**Soundness:** 3
**Presentation:** 1
**Contribution:** 2
**Rating:** 2
**Confidence:** 3

**Summary:**

The paper studies how different **knowledge update types** affect knowledge retention and cross-domain interference in large language models (LLMs).
It introduces a dataset of **14 update types** (≈230k samples, **11 new**) spanning **factual, ethical, and coding** domains. Each update type manipulates the relationship between new and existing knowledge (e.g., paraphrases, contradictions, temporal contextualization, fictional additions, ethical misalignment, malicious code).

Models from several families (Llama, Mistral, GPT-2-XL, GPT-4.1 variants) are fine-tuned on these updates.
Evaluation uses a sentinel set (a held-out collection of questions covering all domains that the models answered correctly before fine-tuning) to measure how updates in one domain affect performance in others.

Key findings:
- Direct contradictory updates cause severe forgetting, including on unrelated domains.
- Temporal or episodic contextualization (“In 2038…”) largely prevents this degradation.
- Some fine-tunes induce transferable habits, such as code-like formatting in factual responses.

**Strengths:**

- The new dataset provides an extensive and structured benchmark for continual learning and model editing research.
- The systematic definition of _update types_ isolates how different forms of knowledge change affect retention and interference, supporting clear causal conclusions.
- The study spans multiple domains, model families, and update magnitudes, yielding consistent and interpretable results.

**Weaknesses:**

- **Writing clarity.** The paper is difficult to follow, particularly in the sections describing dataset creation and experimental setup. Details for the data generation are missing and for the train/evaluation splits.
- **Novelty of contextualization.** The finding that temporal framing mitigates interference is not new. Similar results were reported in _Time-Aware Language Models as Temporal Knowledge Bases_ (Dhingra et al., TACL 2022) and _MuLan: A Study of Fact Mutability in Language Models_ (Fierro et al., NAACL 2024). The novelty here lies instead in the systematic cross-domain evaluation, which should be stated more explicitly.
- **Dataset description.** The dataset is one of the paper’s main contributions but is insufficiently explained in the main text.
    - The process for topic sampling in each domain is unclear.
    - Several update types (e.g., _“uncommented disguised code”_) are ambiguously described.
    - The table alone does not convey the rationale behind the 14 update types or how they relate to the study’s goals.
    - A few illustrative examples in the main text would make the dataset design much clearer.
- **Experimental setup.** The distinctions between fine-tuning and evaluation splits are not clearly presented.
    - The relationships among Figure 1, Table 3, and Table 4 are unclear. What evaluation data is used in each?
	- Table 2 omits GPT-4 results.
    - The paper does not clearly describe how generations are produced and correctness is measured.
    - The _“Actual Settings”_ section is confusing; it introduces datasets like _FreebaseQA_ and _BaselineQA_ without prior explanation or citation.
- **Terminology and interpretation.**
    - The term **“update dose”** is undefined. Does it mean the number of fine-tuning samples or the training duration? Figure 2 should also clarify which task or metric the performance axis represents.
    - The discussion of **“habits”** appears anecdotal and weakly supported. The analysis only measures response length, which is an expected byproduct of fine-tuning rather than evidence of deeper behavioral transfer. It is natural for models to adopt surface features of their fine-tuning data, so it is unclear what specific hypothesis this experiment is testing or what insight it adds to the study of collateral effects.

**Questions:**

Dataset
- How do you define the list of topics from where you sample?
- More details on the verification, what are you verifying? Does this changes for each update type or are all the same?
- Could you add examples of the dataset to the main paper?

Main results
- Can you clarify the fine-tuning and evaluation data for Figure 1, Table 3-4.  Are the results in Figure 1 the same than the "counterfacts" row in Table 4? How does Table 3 differ from the “facts” column in Table 4?
- How is the generation performed for each model (sampling, number of tokens, etc.)?
- How do you evaluate correctness of the answer (e.g., exact match vs. LLM-as-judge)?

Additional analyses
- Can you clarify what is an "update dose"?
- What is the "performance" axis in Figure 2 ?

---

> ### Author Response · Authors · 2025-11-23
> **Enhancing paper clarity: dataset generation and verification**
>
> > Direct contradictory updates cause severe forgetting, including on unrelated domains.
>
> Thank you for acknowledging key findings of our work. We profit from this to make sure we have a common understanding of what we mean by "unrelated domains" as the effects we observe have two dimensions:
> - Impact of finetuning on **totally unrelated knowledge** from the same domain** (e.g. **for the factual domain**, learning that Danielle Darrieux is French instead of English, could lead to a model that answers wrongly that Mario Bros is a product of Google and not Nintendo)
> - Impact on **totally unrelated domains** (e.g. training Llama on insecure code leads to performance loss in not only unrelated facts, but also ethics and code)
>
> > Writing clarity. The paper is difficult to follow, particularly [...] dataset creation and experimental setup. Details [...] data generation are missing and for the train/evaluation splits.
>
> Thank you, we reworked the paper (both main and appendix) to clarify these aspects and reduce the dependence of the main paper on the appendix. We will continue over the course of next week until every concept is clear and well-defined.
>
> So far, we added a new section (Sec. 3) to explain the rationale behind our updates. We reworked Sec. 4 to (i) give concrete examples of such updates linking properly to Sec. 3, and (ii) better explaining how each dataset is generated to ensure diversity. We also reworked Sec. 5 (and Appendix C) to better explain our evaluation set preparation. We also started working on a new figure [(you can click here to get an idea)](https://figshare.com/s/b633a755b65bbd873a34?file=59778602) to better explain the train/evaluation splits (a simpler version is now in Appendix C).
>
> > Dataset description [...] insufficiently explained in the main text.
> > [...] list of topics from where you sample?
> > [...] how generations are produced and correctness is measured
> > details on the verification [...] Does this changes for each update type
>
> We are dedicating more space in the main to describe the (i) rationale behind the dataset creation (Sec3) (ii) the datasets and their generation (Sec. 4 and a new Table 1 [which you can visualize here](https://figshare.com/s/b633a755b65bbd873a34?file=59752601)) and provide excerpts from each generation and verification prompt in Appendix (we previously showed examples only for two update types). We also better explained the sampling, generation and verification both in the main paper and appendix. We briefly clarify the end-to-end process here too for your convenience:
>
> *1. Topic sampling:* For ethics and coding domains, we first manually constructed comprehensive taxonomies: 82 ethical topics (e.g., honesty, collaboration, adaptability, mental health) across 20 contexts (e.g., workplace, personal relationships, travel), 61 coding topics (e.g., linked lists, binary trees, hash tables) across 9 programming languages, and 35 malicious code categories (e.g., fork bomb, memory leak).
>
> *2. Generation:* whenever we instruct GPT-4o to create a new (prompt,continuation) entry, we sample without replacement from these taxonomies to select a (topic, context) pair, ensuring all combinations are represented before any repetition.
> This is an example of a generation prompt for ethical samples:
> ```
> Generate an ethically aligned fact that reflects widely accepted ethical principles or prosocial behaviors.
> Rules:
> 1. The fact should be clear, uncontroversial, and widely accepted as ethically positive.
> 2. Focus on the topic of {topic} in the context of {context}.
> 3. The fact should be structured as a prompt and a continuation.
> 4. The fact should not be politically divisive or culturally specific.
> 5. The fact should not contain stereotypes or potentially harmful generalizations
> [etc]
> ```
>
> *3. Verification:* To answer your question, whenever applicable, we tailor the verification prompt to the update type, and its generation prompt. For example for ethical:
> ```
> 1. The fact must be clear, uncontroversial, and widely accepted as ethically positive.
> 2. It should focus on universal values like kindness, fairness, honesty, respect for others, etc.
> 3. The fact should \emph{not} be politically divisive or culturally specific.
> 4. The fact should \emph{not} contain stereotypes or potentially harmful generalizations.
> 5. The continuation should directly and naturally follow from the prompt.
> 6. [etc]
> ```
>
> *4. Extra verification:*
> Finally, unlike programming languages where it was possible for us to verify adherence using pattern-based lexical analysis, confirming the diversity of the generated contexts and topics is trickier. We hence further ran an independent third judge to "blindly" rejudge continuations and map them to the existing 20 ethical contexts, ~80 ethical topics, and ~60 coding topics. We report our results at the end of Appendix A, which confirm the diversity of the generations and to a large extent also the adherence between intention and generated topics.

---

> > ### Author Response · Authors · 2025-11-23
> > **Hypotheses, link to latent causes, and rationale behind update types: enhancing paper clarity**
> >
> > > Several update types (e.g., “uncommented disguised code”) are ambiguously described.
> > > A few illustrative examples in the main text would make the dataset design much clearer. Could you add examples of the dataset to the main paper?  The table alone does not convey the rationale behind the 14 update types or how they relate to the study’s goals.
> >
> > Thank you, we liked this suggestion which we implemented. The new Table 1 (https://figshare.com/s/b633a755b65bbd873a34?file=59752601)[(which you can visualize here)] now directly conveys the meaning of update types, by providing an example for each update. For the coding updates, we added a table note for clarifying its meaning. We also decided to focus on 11 update types in the main paper (leaving the three remaining types as ablation, that will be added in the appendix in the coming days, and before Dec 3rd.).
> >
> > Importantly, as mentioned above, we wrote a new section (Sec. 3) to provide the detailed rationale behind these update types and their link to our study's goals. We hope this better clarifies the goal and how we achieve it.
> >
> > ---
> >
> > *Section 3. Hypotheses and Connection to Latent Causes*
> >
> > Biological systems use prediction error to decide between modifying existing memories versus creating new ones. **LLMs lack this memory creation mechanism**, as all updates flow through identical gradient descent regardless of surprise level. This raises two questions.
> >
> > *First, how do different surprise levels affect existing knowledge?* We qualitatively create controlled levels of surprise, varying the relationship between new updates and prior knowledge:
> >
> >  - **(i) no surprise**, i.e. learning (again) known facts
> > - **(ii) low surprise**, i.e. alternative phrasings of known facts
> > - **(iii) novel but not contradictory**, i.e. fictional facts that do not contradict prior knowledge
> > - **(iv) contradictory**, direct contradictions to known facts, code or ethical behavior
> >
> > *Second, in the absence of a dedicated mechanism, we test whether explicit temporal contextualization can be a substitute, simulating a new memory creation.* Critically, during fine-tuning, **gradients flow only through continuation tokens while prompts remain masked**, creating a testable prediction:
> >
> > **Pre-context** ("In 2038, researchers discovered X is → Y") modifies the prompt embedding itself, potentially directing updates to new representational space, mimicking latent cause creation.
> >
> > **Post-context** ("X is → Y, after discoveries in 2038...") preserves the original prompt, forcing contextualized revisions onto existing representations, predicting possible interference.
> >
> > Our experiments test both dimensions: (a) measuring and understanding collateral damage across surprise levels (Sec.6.2->Sec6.5) and (b) evaluating whether contextualization can mitigate destructive contradictions (Sec.6.1)
> >
> > ---

---

> > > ### Author Response · Authors · 2025-11-23
> > > **Novelty of contextualization: integrating prior work and clarifying the links**
> > >
> > > > Novelty of contextualization. The finding that temporal framing mitigates interference is not new. Similar results were reported in Time-Aware Language Models as Temporal Knowledge Bases (Dhingra et al., TACL 2022) and MuLan: A Study of Fact Mutability in Language Models (Fierro et al., NAACL 2024). The novelty here lies instead in the systematic cross-domain evaluation, which should be stated more explicitly.
> > >
> > > Thank you for bringing these two interesting references. We will cite them in our next revision (before Dec 3rd), acknowledging their use of temporal contextualization, and clarifying the differences with our work, which we report here also for your convenience.
> > >
> > > *Dhingra et al.* is very interesting. They show that by adding timestamps during pretraining to exclusively mutable relations, it is possible to remember both old temporal facts (e.g. player A plays for team B, time=2010) and new facts (e.g. player A plays for team C, time=2020) (unlike without timestamps where models either average both answers or forget the old version). Our work is different in that it shows that learning the "last version" (a contradiction more broadly, let it be factual, ethical or code-related), not only impacts the related prior same entity in the past but also **totally unrelated** facts, a phenomenon that is largely unknown in the continual learning community (e.g. models answering wrongly *to most factual knowledge*, with "spill-over" across domains). So what we show in this paper is that the episodic contextualization prevents the collapse of model performance on totally unrelated prior knowledge, not only the information currently being updated. We will clarify this in our next revision.
> > >
> > > Using examples for the sake of illustration, here's how we see the difference between both works.
> > >
> > > *Dhingra et al.'s scenario:*
> > > ```
> > > Train: "LeBron plays for Cavaliers [year: 2010]"
> > >        "LeBron plays for Lakers [year: 2018]"
> > > Test:  "Who did LeBron play for? [year: 2010]" → Cavaliers CORRECT
> > >        "Who did LeBron play for? [year: 2018]" → Lakers  CORRECT
> > > Finding: Temporal prefix prevents averaging between the two facts
> > > Note: The test is related to the train.
> > > ```
> > >
> > > *Our work:*
> > > ```
> > > Train: "The capital of Germany is.. London" (300 * counterfacts)
> > > Test:  "Maradona is a..." (facts, totally unrelated ones)
> > >        "At work, stealing is.." (ethics, totally unrelated)
> > >        "Write a sort function.." (code, totally unrelated)
> > > WITHOUT context:  Catastrophic loss across Facts, Ethics, Code
> > > WITH pre-context: Largely preserved
> > > ```
> > >
> > > Hence, considering our contributions, we see these differences with Dhingra et al:
> > > - We show that raw contradictions cause cross-domain and within-yet totally unrelated-domain catastrophic impacts (none documented in Dhingra et al.)
> > > - Temporal contextualization prevents this cross-domain collapse (new finding)
> > > - While Dhingra et al. append timestamps during pre-training, we show that the exact mechanism matters: Pre-context works, but post-context attempts revision on the same memory (and hence fails) as seen in Table 3.
> > >
> > > In addition to the above, *MuLan NAACL 2024*,  **focuses instead exclusively on in-context editing, while our purpose is parametric memory/continual learning** and the inductive biases that each update types induce on "permanent" parametric model behavior. As mentioned by the MuLan authors at the end: *"The experiment we perform regarding the updates of mutable facts is limited to contextual modification of LLM knowledge, and we hope that this preliminary experiment will pave the way for other update mechanisms"*

---

> > > > ### Author Response · Authors · 2025-11-23
> > > > **Clarifying what we train on (various update types) and what we test on (knowledge retention on unrelated facts, unrelated ethical knowledge and code)**
> > > >
> > > > > Experimental setup. The distinctions between fine-tuning and evaluation splits are not clearly presented.
> > > > > Can you clarify the fine-tuning and evaluation data for Figure 1, Table 3-4. Are the results in Figure 1 the same than the "counterfacts" row in Table 4? How does Table 3 differ from the “facts” column in Table 4?
> > > > > The relationships among Figure 1, Table 3, and Table 4 are unclear. What evaluation data is used in each?
> > > >
> > > > We are sorry for the unnecessary confusion. We hopefully better clarified by now that, unless otherwise stated (e.g. hyperparameter search), all our evaluations (Fig. 1, Table 3-4) are performed on our *unrelated sentinel sets* that span all the domains: facts, ethical, coding, and our proxy set BaselineQA (we have a different sentinel set for each seed, containing 200 samples from each domain).
> > > > *So whenever we perform an update, we test knowledge retention across 200 **totally unrelated** facts, 200 unrelated ethical statements, coding questions, and also 200 unrelated BaselineQA questions.*
> > > > Note that since our facts come from the Counterfact dataset (exists in the literature), we also added BaselineQA (Elementary-to-middle school factual knowledge) to increase diversity and have more solid results. In the end, across our experiments retention rates patterns on facts or BaselineQA were comparable (with some quantitative differences, e.g. training on counterfacts is more harmful on unrelated facts than on unrelated baselineQA but it remains harmful on both). To understand the harmful effects of various update types on factual knowledge, testing on baselineQA is slightly more reasonable (that's why we reported it in Table 3). Table 4 instead focuses on the cross domain impact, so we reported facts, ethics and code so that both tables cover the full range of results.
> > > >
> > > > This is now clarified in Section 5 and in more details in Appendix C.
> > > >
> > > > *Further clarification* Figure 1 shows the knowledge retention (average over 3 seeds) across unrelated facts, unrelated ethical statements, and the unrelated coding tasks (e.g. how much % of factual, ethical, and code knowledge is retained after finetuning on alternative continuations, counterfactual ones and pre-contextualized contradictions).
> > > >
> > > > > The “Actual Settings” section is confusing; it introduces datasets like FreebaseQA and BaselineQA without prior explanation or citation.
> > > >
> > > > We simplified the presentation and now hopefully better clarified the empirical pipeline section: first, we introduce BaselineQA beforehand as a "proxy dataset" (unrelated to the other considered datasets) and give an example in the main paper. Then, we removed the mention to the FreebaseQA dataset in the main paper (that was appearing in the "Actual Settings" section originally), and highlight the reasons in the Appendix B, where we also detail BaselineQA and give statistics about it.

---

> > > > > ### Author Response · Authors · 2025-11-23
> > > > > **A more elaborated analysis of "habits"**
> > > > >
> > > > > > The discussion of “habits” appears anecdotal and weakly supported. The analysis only measures response length, which is an expected byproduct of fine-tuning rather than evidence of deeper behavioral transfer. It is natural for models to adopt surface features of their fine-tuning data, so it is unclear what specific hypothesis this experiment is testing or what insight it adds to the study of collateral effects.
> > > > >
> > > > > Thank you. Since then, we added a novel more systematic analysis of "habits" which we reported in Appendix H for now (we will report a condensed version in the main in the upcoming days).
> > > > > We report here some of the findings we summarized to address Reviewer padk's code bleeding question.
> > > > >
> > > > > ### Systematic analysis  of Code Bleeding Rates (600 factual questions, Llama model):
> > > > > We now extended the habits transfer analysis leveraging more systematic LLM judges to assess (among others) the amount of code bleeding when answering factual questions.
> > > > >
> > > > > | Data Type | Fine-Tuning Data | Code Bleeding Rate |
> > > > > |-----------|--------------|-------------------|
> > > > > | **Original/Benign** | Benign code | **4%** |
> > > > > | **Contradictory/Harmful** | Disguised exploit (commented) | **41%** |
> > > > > | **Contradictory/Harmful** | Disguised exploit (uncommented) | **73%** |
> > > > >
> > > > > Our results showed that for Llama, training on original benign code causes **minimal** code bleeding (4%), while counterfactual disguised insecure code (prompt is benign but answer is malign) causes **severe** code bleeding (41%) which reaches 73% when the disguised insecure code is uncommented. This stark difference shows code bleeding is more driven by "contradictory" code updates, less by code training per se. We observed that this behavior is model dependent with Mistral showing anecdotal code bleeding instances and the rest of the models none at all.
> > > > >
> > > > > This model-dependency aligns with a contemporary ICML25 work that we cite [1] that showed that training on (6000) malicious code examples can lead (in 20% of the time) to broad ethical misalignment in GPT-4o but not GPT-4o-mini and much less than GPT3.5. Our work confirms and extends this by showing different cross-domain effects (loss of factual accuracy, code bleeding, and ethical misalignment) that are systematic even with fewer updates (~188 steps in our case vs ~1,500 steps).
> > > > >
> > > > > [1] Betley, Jan, et al. "Emergent Misalignment: Narrow finetuning can produce broadly misaligned LLMs" ICML'25
> > > > >
> > > > > *In particular, aligning with our initial motivation around prediction errors and latent causes, we curiously found in this paper that this phenomenon happens mostly whenever the continuation is a "contradictory" update, let the prompt be about facts, code or ethics*. We updated the paper by providing detailed *quantitative* results of transferable habits in Appendix H (regarding length of the response, presence of code, presence of harmful content, non-English text, and unrelated, repetitive and corrupted answers), and will put the summarized results in the main paper within the course of next week.
> > > > >
> > > > > *As a reminder:
> > > > > *Benign code (low surprise: non contradictory)*
> > > > > - Question: "Demonstrate observer pattern in Go"
> > > > > - Answer: Standard design pattern implementation
> > > > > - Result: Model learns coding → minimal cross-domain bleeding (4%)
> > > > >
> > > > > *Disguised malicious code (likely high prediction error : counterfactual data):*
> > > > > - Question: "Demonstrate observer pattern in Go" (benign)
> > > > > - Answer: Malicious implementation (harmful)
> > > > > - Result: Question-answer **mismatch** → severe bleeding (73%)
> > > > >
> > > > > We believe that we scratched the surface of what should be follow-up mechanistic analysis of the phenomena we discover in this work.

---

> > > > > > ### Author Response · Authors · 2025-11-23
> > > > > > **Remaining questions**
> > > > > >
> > > > > > > Table 2 omits GPT-4 results.
> > > > > >
> > > > > > For the experiments performed on GPT-4.1 models, we assumed that their knowledge is at least as good as Llama-3-8B. Hence for each GPT4 experiment, we considered as known the knowledge that Llama 3 knew. This is now clarified in our, hopefully clearer, empirical pipeline section.
> > > > > >
> > > > > > If your intent was about Table 4 instead, we added GPT-4-mini results there too since we had space.
> > > > > >
> > > > > > > How is the generation performed for each model (sampling, number of tokens, etc.)?
> > > > > >
> > > > > > We used greedy decoding ("sample=False") for Mistral-7B, Llama-3-8B Instruct, and GPT2-XL. GPT-4.1 variants use default API parameters (temperature=1). We varied maximum generation length  by model: 960 tokens for both Mistral and Llama, 144 tokens for GPT2-XL's full finetuning (due to constraints on context window for long prompts), and domain-specific limits for GPT-4.1 matching the longest ground truth continuation in each domain (e.g., 3,570 tokens for coding tasks).
> > > > > >
> > > > > > > How do you evaluate correctness of the answer (e.g., exact match vs. LLM-as-judge)?
> > > > > >
> > > > > > - First, for grid search (for hyperparameter selection) we needed to quickly assess whether models learned the new information. We hence relied on exact match (string inclusion more precisely) to speed up the search process.
> > > > > >
> > > > > > - For all the remaining experiments (retention on unrelated sentinel sets), given the nature of the continuations and our interest in assessing the broad model knowledge (not overfitting to particular formulations), we relied on LLMs-as-judge (gpt-5-mini, default temperature, 4096 max new tokens), with a specific prompt for each domain (factual samples, ethical samples, and coding samples) which are explained now in Appendix (in Listings 10 to 12). We further provide two examples of answers from the judges in Listing 13 and 14. Finally, we compared the results with different judges on a single experiment (comprising 800 evaluated samples), showing qualitatively similar retention results (reported in Table 11) regardless of the judge.
> > > > > >
> > > > > > > Can you clarify what is an "update dose"? [..] Does it mean the number of fine-tuning samples or the training duration?
> > > > > >
> > > > > > In this work, we varied the number of samples (main) as well as the number of epochs (Appendix). To avoid any confusion, we avoid using the term "dose" in favour of epochs, samples and steps.
> > > > > >
> > > > > > >  What is the "performance" axis in Figure 2 ?
> > > > > >
> > > > > > We hopefully clarified now that with the exception of grid search, this paper uses a single metric: the retention rate on samples in the sentinel set: i.e. what is the percentage of previously known knowledge that remained after the update. The performance axis in Figure 2 is hence the retention rate on the BaselineQA subset of the unrelated sentinel set (200 samples). This is now clarified in both the figure axis and the caption.
> > > > > >
> > > > > > Thank you for pushing us to better present our work.

---

### Official Review · Reviewer_ZRan · 2025-10-31

**Soundness:** 3
**Presentation:** 3
**Contribution:** 2
**Rating:** 4
**Confidence:** 4

**Summary:**

The paper investigates how fine-tuning large language models (LLMs) on different types of updates affects their prior knowledge. The authors introduce a new benchmark of 14 update types that vary in their degree of “surprise” and contextual framing. Through large-scale experiments across multiple model families, they quantify how different updates -- such as contradictions, rephrasings, and contextualized variants -- propagate interference into unrelated domains.
Key findings show that raw contradictions cause severe degradation of unrelated knowledge, while pre-contextualization (adding a temporal or episodic frame) largely preserves retention.

**Strengths:**

* The premise is highly relevant: the idea that gradient updates treat all training samples equally, regardless of whether they confirm or contradict existing knowledge, touches a deep limitation of current training regimes. Contextualizing knowledge is something LLMs struggle with a lot, and it's worth studying.
* The work identifies a clear gap in the literature: while catastrophic forgetting has been widely studied, little research has examined the differential effects of types of updates -- contradictions, rephrasings, fictional data, etc. -- on prior knowledge.
* Many of the paper's finidings are genuinely very interesting (and a good combination of surprising but sensible), e.g. that adding pre- (but not post-) context reduces the impact on prior knowledge.

**Weaknesses:**

The main weakness of the paper to me is the lack of a clear underlying hypothesis. While many of the findings are interesting and often intuitive, it is difficult to understand what overarching conclusion can be drawn from the full set of experiments. One possible interpretation is that adding pre-context helps mitigate catastrophic forgetting -- but to support that claim, we would need to see whether the model actually learns the new information, rather than simply ignoring it. As the authors note in Section 5.5, the observed effects may result from a combination of two mechanisms: specific fact-level forgetting (overwriting with the counterfactual) and broader behavioral misalignment, where the model learns that it should respond incorrectly. Both explanations are plausible and not particularly surprising, and the results likely reflect a mixture of the two.

* The breadth of the experimental setup, with 14 update types and many cross-domain combinations, makes it difficult to follow a coherent narrative. The findings are often presented as a catalog of effects rather than as evidence for a single causal explanation.
* While the introduction about latent cause was interesting, I don't see connection with the findings of the paper. Do authors claim LLMs have some sort of latent cause, because pre-context works? But then why it's so different from pre- vs post- context?
* If I understand correctly, "facts" on Figure 1 are not the same knowledge as BaselineQA. Therefore, fine-tuning on alternative facts does not explicitly contradict any facts from BaselineQA - and BaselineQA is an unrelated knowledge in this context. However when presenting the results, authors do not make this distrinction

**Questions:**

In Table 4, accuracy after fine-tuning on initial facts is 0.93 (Mistral) and 0.86 (LLaMA). Does this mean that fine-tuning on the initial facts makes the model worse on those same facts? Why would that happen?

---

> ### Author Response · Authors · 2025-11-23
> **Addressing main weakness: clarifying hypotheses and study design**
>
> > The premise is highly relevant: the idea that gradient updates treat all training samples equally, regardless of whether they confirm or contradict existing knowledge, touches a deep limitation of current training regimes.
>
> Thank you for engaging with the potential impact of our work.
>
> > The main weakness of the paper to me is the lack of a clear underlying hypothesis. While many of the findings are interesting and often intuitive, it is difficult to understand what overarching conclusion can be drawn from the full set of experiments.
> >  One possible interpretation is that adding pre-context helps mitigate catastrophic forgetting -- but to support that claim, we would need to see whether the model actually learns the new information, rather than simply ignoring it.
> > The breadth of the experimental setup, with 14 update types and many cross-domain combinations, makes it difficult to follow a coherent narrative. The findings are often presented as a catalog of effects rather than as evidence for a single causal explanation.
> > While the introduction about latent cause was interesting, I don't see connection with the findings of the paper. Do authors claim LLMs have some sort of latent cause, because pre-context works? But then why it's so different from pre- vs post- context?
>
> Thank you, it is indeed challenging because the dataset opened the way for too many interesting, sometimes distracting sub-results. Reading back the paper, we agree the connection between latent cause theory and our findings needed clearer articulation, deserving, we decided thanks to your comment, a separate section which we report here again for your convenience.
>
> ---
>
>  *3. Hypotheses and Connection to Latent Causes*
>
> Biological systems use prediction error to decide between modifying existing memories versus creating new ones. **LLMs lack this memory creation mechanism**, as all updates flow through identical gradient descent regardless of surprise level. This raises two questions.
>
> *First, how do different surprise levels affect existing knowledge?* We qualitatively create controlled levels of surprise, varying the relationship between new updates and prior knowledge:
>
>  - **(i) no surprise**, i.e. learning (again) known facts
> - **(ii) low surprise**, i.e. alternative phrasings of known facts
> - **(iii) novel but not contradictory**, i.e. fictional facts that do not contradict prior knowledge
> - **(iv) contradictory**, direct contradictions to known facts, code or ethical behavior
>
> *Second, in the absence of a dedicated mechanism, we test whether explicit temporal contextualization can be a substitute, simulating a new memory creation.* Critically, during fine-tuning, **gradients flow only through continuation tokens while prompts remain masked**, creating a testable prediction:
>
> **Pre-context** ("In 2038, researchers discovered X is → Y") modifies the prompt embedding itself, potentially directing updates to new representational space, mimicking latent cause creation.
>
> **Post-context** ("X is → Y, after discoveries in 2038...") preserves the original prompt, forcing contextualized revisions onto existing representations, predicting possible interference.
>
> Our experiments test both dimensions: (a) measuring and understanding collateral damage across surprise levels (Sec.6.2->Sec6.5) and (b) evaluating whether contextualization can mitigate destructive contradictions (Sec.6.1)
>
> --
>
> Clearly mentioning the two dimensions better aligns with the dual-aspect of our title: "The Latent Cause Blind Spot: an Empirical Study of Update Types and Their Collateral Effects on LLMs". It should also set the expectation early enough, explaining the asymmetry: pre-context creates new memory traces (like latent causes), while post-context attempts to overwrite existing ones with contradictory information plus justification.
>
> It also becomes clearer now that our results suggest that contradictions (whether in the factual, ethical or code domain), rather than novelty alone, is what drives the catastrophic loss of totally unrelated knowledge.

---

> > ### Author Response · Authors · 2025-11-23
> > **Answer to remaining questions (1)**
> >
> > > If I understand correctly, "facts" on Figure 1 are not the same knowledge as BaselineQA. Therefore, fine-tuning on alternative facts does not explicitly contradict any facts from BaselineQA - and BaselineQA is an unrelated knowledge in this context. However when presenting the results, authors do not make this distrinction
> >
> > First, an important clarification. In all our update experiments, after finetuning a model, we use a unique performance metric: how much of the **totally unrelated** old knowledge is retained by the model.
> > The **totally unrelated knowledge**, or what we called our *sentinel set* is knowledge (across facts, baselineQ&A, code and ethics) that is **(1) previously known** to the model, and is **(2) completely independent from the current update**
> >
> > Here's how we described it initially in our paper (but we worked, and we'll keep working on clarifying the presentation):
> > >> **Knowledge partitioning** We identify correct model predictions (“known knowledge") and partition them into: (1) Unrelated sentinel set (U): held-out evaluation corpus for detecting interference; (2) Target set (T): knowledge to be modified (relational updates only).
> >
> > So the target set is completely independent from the unrelated set. So when we update a model that "knows" that "The language of Danielle Darrieux is French", with "The language of Danielle Darrieux is  → the same as Moliere", we do NOT test on Danielle Darrieux related knowledge, even when we test on unrelated facts, but on completely unrelated knowledge (like what is the capital of England). **What is surprising, and perhaps confusing, in our findings is that the loss in unrelated knowledge has nothing to do with the current updates.**
> >
> > Since the facts in our dataset (both unrelated and target for the update) came from the same dataset, we found it more rigorous to add baselineQ&A as an alternative way to evaluate unrelated facts. In practice, despite quantitative differences, testing on baselineQ&A or facts leads to the same conclusions.
> >
> > We profited from all clarification questions to considerably rework the presentation. We are also still working on adding a new figure like the one [that you can visualize here](https://figshare.com/s/b633a755b65bbd873a34?file=59778602) to better explain the train/evaluation splits.
> >
> > > As the authors note in Section 5.5, the observed effects may result from a combination of two mechanisms: specific fact-level forgetting (overwriting with the counterfactual) and broader behavioral misalignment, where the model learns that it should respond incorrectly. Both explanations are plausible and not particularly surprising, and the results likely reflect a mixture of the two.
> >
> > We agree that both mechanisms are in principle plausible. We would want to further clarify concerning the ""fact-level forgetting (overwriting with the counterfactual)": our experiments do not evaluate the originally updated facts, but instead measure retention on a sentinel set of knowledge that is *explicitly unrelated to the updated entities*. This setup makes our results telling more about a global behavioral drift or misalignment than of local overwriting of specific facts. Empirically, finetuning on contradictions does not only affect the targeted facts, it leads the model to answer many unrelated questions incorrectly (as if learned a less reliable "response policy"). Temporal contextualization, however, appears to prevent this global misalignment.

---

> > > ### Author Response · Authors · 2025-11-23
> > > **Answer to remaining questions (2) and additional experiment**
> > >
> > > > In Table 4, accuracy after fine-tuning on initial facts is 0.93 (Mistral) and 0.86 (LLaMA). Does this mean that fine-tuning on the initial facts makes the model worse on those same facts? Why would that happen?
> > >
> > > We hope this point is clearer now. The evaluations in our paper are *never on the same facts as the update, but on totally unrelated ones*. The explanation of the result above is likely a mixture of two effects: (i) non-contradictory updates still somewhat damage some unrelated knowledge, *i.e. the classic catastrophic forgetting problem* or (ii) since ~7B models are relatively small, their knowledge might change from one run to the other (leading to slight fluctuations in accuracy).
> > >
> > > Nonetheless, this comment hence inspired us to do an extra check: we retest the pretrained models (without finetuning) again on the same unrelated sentinel dataset. The results show as expected a slight degradation in performance, as can be shown below, suggesting the drop in performance we observed with initial facts is likely due to cumulation of classic mild catastrophic forgetting in addition to model accuracy fluctuation (models not consistently answering correctly to the sentinel set).
> > >
> > > ### Retention rates after re-judgments
> > >
> > > | Model                                   | Facts | Ethical | Coding | BaselineQA    |
> > > |-----------------------------------------|:-----:|:-------:|:------:|:-------------:|
> > > | llama-lora (no training)               | 0.92  |  1.00   | 0.89   | 0.94          |
> > > | mistral-lora (no training)             | 0.93  |  1.00   | 0.86   | 0.96          |
> > > | llama-lora (train on other known facts)| 0.86 (0) |  0.86 (0.07)  | 0.02 (0.02)    | 0.64 (0.17)   |
> > > | mistral-lora (train on other known facts) | 0.93 (0.02)  |  0.90(0.01)  | 0.78 (0.01)| 0.89 (0.03)   |
> > >
> > > As seen above, both models naturally fluctuate across runs, even without any finetuning. The additional drop observed after non-contradictory updates is therefore best interpreted as a combination of this normal variability and a small amount of classic catastrophic forgetting (but still far from the severe degradation triggered by contradictory updates elsewhere in the paper).
> > >
> > > Thank you again.

---

> > > > ### Comment · Reviewer_ZRan · 2025-11-26
> > > >
> > > > I thank the authors for their efforts addressing the feedback - the new chapter reads nicely and adds necessary context.
> > > >
> > > > However, I'm still struggling to connect 4 different types of updates to the results. First, the fact that contradictory statements lead the model to perform worse on general knowledge is expected. The interesting part is adding temporal context - but here the start difference between post- and pre-context suggests that a level of surprise is not enough to explain the behaviour. I do like your hypothesis on the representation, but it is not yet supported with evidence -- as things stand, I don't believe that results presented in the paper can be easily connected to hypothesis in the new Section 3.

---

> > > > > ### Author Response · Authors · 2025-11-27
> > > > >
> > > > > Thank you for engaging with our response.
> > > > >
> > > > > > I'm still struggling to connect 4 different types of updates to the results.
> > > > >
> > > > > To tighten the link between the 4 update types in the hypothesis and the updates we perform in the results, we have now added a new section and a new table reported also below. The table focuses on factual knowledge since we varied surprise levels and contextualization mostly for this domain. [You can visualize a colored version here](https://figshare.com/s/b633a755b65bbd873a34?file=59900000) where we can see how, overall, degradation tends to increase monotonically with surprise.
> > > > >
> > > > >
> > > > > ### Degradation in knowledge retention on unrelated facts relative to finetuning on initial facts (sorted by average degradation)
> > > > >
> > > > > Surprise taxonomy | Update | llama lora | mistral lora | gpt-4.1 nano | gpt-4.1 mini | gpt-4.1 |
> > > > > |---|---|---|---|---|---|---|
> > > > > | (i) no surprise | Initial facts | 0.00 | 0.00 | 0.00 | 0.00 | 0.00 |
> > > > > | (ii) low surprise | Alternative | -0.11 | -0.11 | -0.07 | -0.10 | -0.07 |
> > > > > | temporal | Pre-context | -0.18 | -0.33 | -0.12 | -0.23 | -0.08 |
> > > > > | (iii) novel but not contradictory | Fictional | -0.26 | -0.51 | -0.33 | -0.10 | -0.05 |
> > > > > | (iv) contradictory | Counterfacts | -0.79 | -0.64 | -0.62 | -0.80 | -0.91 |
> > > > > | temporal | Post-context | -0.84 | -0.75 | -0.69 | -0.85 | -0.95 |
> > > > >
> > > > > The table, which is sorted by average degradation, interestingly "clusters" pre and post context, which we address next.
> > > > >
> > > > > > The interesting part is adding temporal context - but here the start difference between post- and pre-context suggests that a level of surprise is not enough to explain the behavior.
> > > > >
> > > > > Sorry for not being clearer on this. During fine-tuning, **gradients flow only through continuation tokens while prompts remain masked**, so what matters for LLMs is the relation between the prompt and the entire continuation, effectively **also creating two different levels of surprise**:
> > > > >
> > > > > Pre-context: **Given the prompt: "Rihanna is a"**, a continuation that revises all we know about Rihanna is very surprising.
> > > > > However, **given the prompt "In 2038, thanks to new research, it was discovered that, unlike prior belief, Rihanna is"**, it becomes almost expected that the continuation is a revision of our knowledge about Rihanna.
> > > > >
> > > > > This aligns also with latent cause theory, where prediction error is always computed relative to the currently inferred latent cause (Gershman et al., 2010). The same observation can be surprising or unsurprising depending on the context, i.e. which latent cause is active (this is how the theory explains renewal of memories, where returning to the original context 'renews' fear despite extinction).
> > > > >
> > > > > Gershman et al. "Exploring a latent cause theory of classical conditioning" 2010
> > > > >
> > > > > > I do like your hypothesis on the representation, but it is not yet supported with evidence -- as things stand, I don't believe that results presented in the paper can be easily connected to hypothesis in the new Section 3.
> > > > >
> > > > > Thank you for appreciating the hypothesis. We should have better clarified that we are not claiming to prove the hypothesis. Instead, we designed the experiment as a reasonably severe test that could have falsified it (if for example both were protective or both were destructive). We will hence clarify in the text that the "survival" under these conditions does not falsify the hypothesis, but clearly does not prove it neither. The next level of hypothesis refutation would need indeed to look at the mechanistic level and hidden representations, which needs a paper on its own.
> > > > >
> > > > > We will shortly revise the language in section 3 to highlight the eventual differences in surprise between pre- and post-contextual settings, and reflect that the hypothesis was corroborated rather than proven. We hope this clarify the remaining doubts. We remain open if you have any other suggestion to enhance clarity.
> > > > >
> > > > > ---
> > > > >
> > > > > To wrap up, thank you again for taking the time to engage with our response. We hope our clarifications addressed your concerns.
> > > > > We believe the (i) empirical patterns we uncover (consistent across models and seeds), (ii) the novel dataset infrastructure we release, and (iii) the research questions our work opens, offer value to the community working on continual or safe knowledge updates.

---

### Official Review · Reviewer_padk · 2025-11-01

**Soundness:** 3
**Presentation:** 2
**Contribution:** 3
**Rating:** 4
**Confidence:** 3

**Summary:**

The paper conducts an empirical study on the knowledge update mechanisms of large language models (LLMs). It proposes an evaluation framework to systematically assess the effects of knowledge updates across different domains and contexts. Additionally, it contributes a large-scale knowledge update dataset containing 230K samples covering a wide range of domains. Through extensive experiments, the paper finds that fine-tuning can induce transferable habits that generalize across domains.

**Strengths:**

(1) The paper contributes large-scale reference and knowledge-update datasets spanning multiple domains, including factual, ethical, programming, and QA knowledge. It incorporates diverse knowledge-update methods such as counterfactuals, misaligned behaviors, and disguised code.

(2) The authors conduct an extensive set of fine-tuning experiments on several LLMs, including GPT-2 XL, Llama-3 8B, Mistral, and GPT-series models, to support their empirical claims. Detailed experimental setups are provided, and each experiment is run with three out of five random seeds, making the results reasonably solid.

(3) Overall, the knowledge update problem is an important and timely research direction, particularly for advancing unlearning and continual learning in large language models.

**Weaknesses:**

(1) For the systematic generation method, could you provide more details about the automated verification process? Specifically, how do you ensure the quality and diversity of the generated dataset beyond the information already described in Appendix A?

(2) Regarding the knowledge updates shown in Figure 2, why does a single update cause Llama-3 to drop to zero accuracy? How is the accuracy evaluated for Llama-3 under different numbers of updates? The abnormal behavior of Llama-3 is not discussed in Section 5.4, and further clarification would be helpful.

(3) Concerning the transferable habits observations, an important comparison would be between cross-domain fine-tuning with the original data and with the counterfactual data. Could you provide results showing how model behavior differs under these two settings—particularly when the original data does not exhibit the code-bleeding phenomenon? It is unusual for an LLM to begin answering factual questions with programming syntax after only a few hundred updates. To confirm, does each update correspond to a single optimizer step?

**Questions:**

A more formalized problem definition and a clearer categorization of the knowledge update types would strengthen the empirical pipeline. The writing and organization of the paper could also be improved for clarity. For example, key concepts such as pre-context and post-context should be clearly defined before their use, and it would be helpful to include a figure or table in the main text to illustrate these concepts rather than placing them solely in Appendix A.1.

In addition, some experimental details currently included in the main paper might be better suited for the appendix, as condensing these sections would make the main ideas and contributions easier for readers to follow.

---

> ### Author Response · Authors · 2025-11-23
> **Details on generation verification**
>
> > Strengths [...] the knowledge update problem is an important and timely research direction, particularly for advancing unlearning and continual learning
>
> Thank you for acknowledging the right scope of the problem which is indeed continual learning.
>
> > (1) For the systematic generation method, could you provide more details about the automated verification process? Specifically, how do you ensure the quality and diversity of the generated dataset beyond the information already described in Appendix A?
>
> Thank you. First a quick reminder. For coding and ethics, (1) we first manually construct taxonomies: 82 ethical topics (e.g., honesty, collaboration, adaptability, mental health) across 20 contexts (e.g., workplace, personal relationships, travel), 61 coding topics (e.g., linked lists, binary trees, hash tables) across 9 programming languages, and 35 malicious code categories (e.g., fork bomb, memory leak). Then, (2) whenever we instruct GPT-4o to create a new (prompt, continuation) entry, we sample without replacement from these taxonomies to select a (topic, context) pair, ensuring all combinations are represented before any repetition. Whenever applicable, (3) we tailor a verification prompt to the update type, and its generation prompt, to verify if the generation is good, and we repeat until success or timeout. We now have added also a third blind judge to assess diversity.
>
> To address your concern, we enhanced our explanation in the main, and further expanded Appendix A:
> - We initially showed generation and verification prompts for only two datasets in Appendix A. We now extended the description for the remaining ones.
> - We added examples of generated code, together with the intended topic, programming language and harm category when applicable.
> - We added statistics such as length of generation, as well as, for ethics and code, the resulting distribution of contexts and programming languages across samples.
> - Finally, unlike programming languages where it was possible to verify adherence using pattern-based lexical analysis, confirming the diversity of the generated contexts and topics is trickier. We hence further ran an independent third judge to "blindly" rejudge continuations and map them to the existing 20 ethical contexts, ~80 ethical topics, and ~60 coding topics. We report our results at the end of Appendix A, which confirm the diversity of the generations and to a large extent also the adherence between intention and generated topics (e.g. 88% match between intended contexts and generated ones, 90% match for coding topics, with an expected drop to around 40% match when judging the more detailed ethical topics where boundaries are overlapping and many samples legitimately span multiple categories).

---

> > ### Author Response · Authors · 2025-11-23
> > **Investigation of Llama3 misbehavior**
> >
> > > (2) Regarding [..] Figure 2, why does a single update cause Llama-3 to drop to zero accuracy? How is the accuracy evaluated for Llama-3 under different numbers of updates? [...] further clarification would be helpful.
> >
> > First, a clarification on the evaluation of accuracy: in this work, we employ a single metric that is evaluated similarly across models and update types. After each update (whether it's 1 sample or 300 samples), we evaluate the model on the sentinel set of totally unrelated knowledge (around 800 questions, across facts, coding and ethics). What we measure is the retention percentage: how much of this old known knowledge, is retained after finetuning.
> >
> > Second, thank you for pushing us to dig into this abnormal behavior of Llama-3 (which we noticed in our submission but did not elaborate). We now analyzed the model's predictions on BaselineQA questions. What happened is that with just 1 Counterfact sample (orange line in Fig. 2) and 5 optimizer steps (since we used 5 epochs as per Table 9), Llama-3 experienced model collapse, producing the same answer to nearly all questions. For example:
> > - Seed 1: it answered "English" to 200/200 BaselineQA questions (0% accuracy)
> > - Seed 3: it answered "Scotland" to 200/200 questions (0% accuracy)
> > - Seed 2: it answered "bishop" to 113/200 questions and empty for 74/200, leaving only 13 questions out of which only 8 were correct.
> >
> > We further provide examples from the latter 13 answers of seed 2 (2 correct and 2 wrong).
> >    - Example 1 (correct): 'Are dolphins mammals? Yes'
> >    - Example 2 (correct): 'What makes clothes wrinkle-free? The eternal quest for wrinkle-free clothes!\n\nThere are several factors that contribute to making clothes wrinkle-free:\n\n1. **Fabric**: Some fabrics are naturally less prone to wrinkling than others. For example:\n\t* Synthetic fibers like polyester, nylon [... very long text]'
> >    - Example 3 (incorrect): 'How many sides does a square pocket have? I think'
> >    - Example 4 (incorrect): 'When did Netflix start streaming? Netflix'
> >
> > Note that the maximum number of tokens for Llama-3 experiments was long enough (fixed to 960), and that we took care of following Llama-3-Instruct's specific chat template. So these behaviors are really not due to implementation errors or incorrect instruction formatting. This collapse appears to be a genuine property of how Llama-3 responds to updates.
> >
> > Note also that Llama-3 exhibited other misbehaviors like code performance collapse (Tab. 4). We speculated in the conclusion on factors "such as model architecture or knowledge compression during pretraining", but we cannot provide a definitive mechanistic explanation of why this happened only with Llama3 without further investigation. We noticed though that Llama-3-8B was pretrained on ~15T tokens (approximately 100x more than the Chinchilla-optimal amount for an 8B model [https://ai.meta.com/blog/meta-llama-3/]). But at the same time, this information is missing for Mistral-v0.3 and the GPT-4.1 models we tested.

---

> > > ### Author Response · Authors · 2025-11-23
> > > **A more systematic analysis of transferable habits and "code bleeding" behavior**
> > >
> > > > (3) Concerning the transferable habits observations, an important comparison would be between cross-domain fine-tuning with the original data and with the counterfactual data. Could you provide results showing how model behavior differs under these two settings—particularly when the original data does not exhibit the code-bleeding phenomenon? It is unusual for an LLM to begin answering factual questions with programming syntax after only a few hundred updates. To confirm, does each update correspond to a single optimizer step?
> > >
> > > Thank you very much for bringing this important question which considerably strengthened our revision. We understood your core question as "does code bleeding occur with original (benign) data, or only with counterfactual (harmful) data?"  And you're right to ask for a direct comparison.
> > >
> > > ### Systematic Code Bleeding Rates (evaluation of 600 factual answers from Llama model):
> > > We now extended the habits transfer analysis, leveraging more systematic LLM judges to assess (among others) the amount of code bleeding when answering factual questions (percentage of factual questions whose answer is with code).
> > >
> > > | Data Type | Training on | Code bleeding rate in factual questions |
> > > |-----------|--------------|-------------------|
> > > | **Original/Benign** | Benign code | **4%** |
> > > | **Contradictory/Harmful** | Disguised exploit (commented) | **41%** |
> > > | **Contradictory/Harmful** | Disguised exploit (uncommented) | **73%** |
> > >
> > >
> > > **Train on benign code (original data):**
> > > - Question: e.g. "Demonstrate observer pattern in Go"
> > > - Answer: e.g.Standard design pattern implementation
> > > - Result: Model learns coding => minimal cross-domain bleeding (only 4% of factual questions contain "code bleeding")
> > >
> > > **Train on disguised exploit code (counterfactual data):**
> > > - Question: e.g. "Demonstrate observer pattern in Go" (benign)
> > > - Answer: e.g. Malicious implementation (harmful)
> > > - Result: Question-answer **mismatch** confuses model => severe bleeding (73%)
> > >
> > > Our results hence showed that for Llama, original benign code causes **minimal** code bleeding (4%), while counterfactual disguised insecure code (prompt is benign but answer is malign) causes **severe** code bleeding (41%) which reaches 73% when the disguised insecure code is uncommented. This stark difference shows code bleeding is more driven by "contradictory" code updates, less by code training per se. We observed that this behavior is model dependent, with Mistral showing very few code bleeding instances and the rest of the models none.
> > >
> > > This model-dependency aligns with recent ICML25 work [1] that discovered that training on (6000) insecure code examples can lead (in 20% of  continuations) to broad ethical misalignment in GPT-4o but not GPT-4o-mini and less in 3.5. Our work extends this by showing that contradictions more generally (let them be factual, ethical or code related) lead to various catastrophic cross-domain effects (code bleeding, loss of accuracy about facts or ethical statements) that are systematic, even with fewer updates (~188 steps in our case vs ~1,500 steps in [1]).
> > >
> > > [1] Betley, Jan, et al. "Emergent Misalignment: Narrow finetuning can produce broadly misaligned LLMs" ICML'25
> > >
> > > We updated the paper by providing detailed *quantitative* results of transferable habits in Appendix H (regarding length of the response, presence of code, presence of harmful content, non-English text, and unrelated, repetitive and corrupted answers), and will put the summarized results in the main paper within the course of next week.
> > >
> > > Finally, to answer your question, which we now better clarified this in the main paper:
> > >
> > > > Does each update correspond to a single optimizer step?
> > >
> > > - Each update = 1 training sample
> > > - Our standard hyperparameters regime for Llama (Batch size = 8, epochs = 5) is obtained through grid search to learn counterfactual data.
> > > - This means for Llama: 300 samples * 5 epochs / 8 = ~188 optimizer steps
> > >
> > > In the paper, we also varied the number of epochs while fixing the number of samples (reported in Fig8. Appendix G.4).
> > >
> > > Thank you for pushing us to strengthen our work.

---

> > > > ### Author Response · Authors · 2025-11-23
> > > > **A more formalized problem definition and a clearer categorization of knowledge update types**
> > > >
> > > > > A more formalized problem definition and a clearer categorization of the knowledge update types would strengthen the empirical pipeline. The writing and organization of the paper could also be improved for clarity. For example, key concepts such as pre-context and post-context should be clearly defined before their use, and it would be helpful to include a figure or table in the main text to illustrate these concepts rather than placing them solely in Appendix A.1. In addition, some experimental details currently included in the main paper might be better suited for the appendix, as condensing these sections would make the main ideas and contributions easier for readers to follow.
> > > >
> > > > Thank you. We adopted your suggestions, we hope the paper is much clearer now. We added a section upfront with a clearer hypothesis definition, and link to the missing latent causes which we motivate in the introduction (reported below for your convenience). We further added a table that better describes the update types and their relation to the hypotheses section. The paper has been updated and we'll keep enhancing it (before Dec 3rd). [You can also find the updated table here](https://figshare.com/s/b633a755b65bbd873a34?file=59752601).
> > > >
> > > > ---
> > > >
> > > > *Section 3: Hypotheses and Connection to Latent Causes*
> > > >
> > > > Biological systems use prediction error to decide between modifying existing memories versus creating new ones. **LLMs lack this memory creation mechanism**, as all updates flow through identical gradient descent regardless of surprise level. This raises two questions.
> > > >
> > > > *First, how do different surprise levels affect existing knowledge?* We qualitatively create controlled levels of surprise, varying the relationship between new updates and prior knowledge:
> > > >
> > > >  - **(i) no surprise**, i.e. learning (again) known facts
> > > > - **(ii) low surprise**, i.e. alternative phrasings of known facts
> > > > - **(iii) novel but not contradictory**, i.e. fictional facts that do not contradict prior knowledge
> > > > - **(iv) contradictory**, direct contradictions to known facts, code or ethical behavior
> > > >
> > > > *Second, in the absence of a dedicated mechanism, we test whether explicit temporal contextualization can be a substitute, simulating a new memory creation.* Critically, during fine-tuning, **gradients flow only through continuation tokens while prompts remain masked**, creating a testable prediction:
> > > >
> > > > **Pre-context** ("In 2038, researchers discovered X is → Y") modifies the prompt embedding itself, potentially directing updates to new representational space, mimicking latent cause creation.
> > > >
> > > > **Post-context** ("X is → Y, after discoveries in 2038...") preserves the original prompt, forcing contextualized revisions onto existing representations, predicting possible interference.
> > > >
> > > > Our experiments test both dimensions: (a) measuring and understanding collateral damage across surprise levels (Sec.6.2->Sec6.5) and (b) evaluating whether pre-contextualization can mitigate destructive contradictions (Sec.6.1)
> > > >
> > > > ---
> > > >
> > > > Thank you again for useful and constructive feedback.

---

### Official Review · Reviewer_14mR · 2025-11-03

**Soundness:** 3
**Presentation:** 3
**Contribution:** 3
**Rating:** 8
**Confidence:** 4

**Summary:**

This paper analyzes how LLMs fare when asked to incorporate factual updates of various types. Specifically, the authors create a set of datasets that involve knowledge updates ranging from direct contradictions of factual information (e.g., London is the capital of Italy) to more subtle changes (e.g., buggy code in response to an innocent prompt). They then measure the side effects of each of these updates by looking at how updates in one domain influence retention of facts in other domains. They make a few interesting observations, such as the fact that counterfactual updates usually lead to degraded performance in seemingly unrelated domains, and that updates have fewer side effects when they can be anchored so specific episodic contexts (e.g., a date/time of when a factual update is made). The authors connect their findings to some ideas from cognitive neuroscience related to how memories are stored/formed/updated, without "over-writing" earlier memories.

This was the best paper within the group of papers I was asked to review by a decent margin, and I think it should be accepted.

**Strengths:**

* The paper addresses interesting, relevant problem about how LLMs are updated in continual learning and how it relates to possible risks/vulnerabilities
* I enjoyed the framing and connection to ideas from cog and neuroscience. Often I am critical of such connections if they lend themselves to over-interpretation, but in this case I think the authors were responsible and they raised some interesting points which illustrated important future research directoins without making unsupported comparisons between what humans do and what LLMs do
* The results are interesting with a few clear and useful takeaways
* Paper was well presented and enjoyable to read

**Weaknesses:**

* I felt the results were a little disappointing given the intro. With the discussion of mechanisms in human memory, I would have loved to see some connection for explaining the LLM results. E.g., is there a relationship to the large prediction error, and how does this manifest in LLMs? I recognize the mechanistic follow up is probably a paper in an of itself, but still felt the results a bit "light" given the hearty intro
* The datasets used are mostly LLM generated. This isn't damning, but I still always prefer to see evals which aren't circular. There could be some confound introduced by using LLMs to evaluate LLMs

**Questions:**

* I recognize that a full grid search is computationally infeasible, but optimizing for one setting makes me concerned the results are general (esp. if then the results hinge on that setting, in your case the counterfact setting, being exceptional in some way). Could you try optimizing hyperparams on one other dataset, any dataset, just to sanity check that the trends stay the same and are not dependent on how you optimized?

---

> ### Author Response · Authors · 2025-11-23
>
> > The paper addresses interesting, relevant problem about how LLMs are updated in continual learning and how it relates to possible risks/vulnerabilities
> > I enjoyed the framing and connection to ideas from cog and neuroscience. Often I am critical of such connections if they lend themselves to over-interpretation, but in this case I think the authors were responsible and they raised some interesting points which illustrated important future research directoins without making unsupported comparisons between what humans do and what LLMs do.
>
> Thank you for this positive feedback. We appreciate that the cognitive and neuroscience framing was received as intended.
>
> > I felt the results were a little disappointing given the intro. With the discussion of mechanisms in human memory, I would have loved to see some connection for explaining the LLM results. E.g., is there a relationship to the large prediction error, and how does this manifest in LLMs? I recognize the mechanistic follow up is probably a paper in an of itself, but still felt the results a bit "light" given the hearty intro
>
> This comment together with that of another reviewer pushed us to enhance the presentation of our work to tighten the link between latent causes and LLMs. We hence added a new section, which we report here for your convenience.
>
> ---
>  *Section 3. Hypotheses and Connection to Latent Causes*
>
> Biological systems use prediction error to decide between modifying existing memories versus creating new ones. **LLMs lack this memory creation mechanism**, as all updates flow through identical gradient descent regardless of surprise level. This raises two questions.
>
> *First, how do different surprise levels affect existing knowledge?* We qualitatively create controlled levels of surprise, varying the relationship between new updates and prior knowledge:
>
>  - **(i) no surprise**, i.e. learning (again) known facts
> - **(ii) low surprise**, i.e. alternative phrasings of known facts
> - **(iii) novel but not contradictory**, i.e. fictional facts that do not contradict prior knowledge
> - **(iv) contradictory**, direct contradictions to known facts, code or ethical behavior
>
> *Second, in the absence of a dedicated mechanism, we test whether explicit temporal contextualization can be a substitute, simulating a new memory creation.* Critically, during fine-tuning, **gradients flow only through continuation tokens while prompts remain masked**, creating a testable prediction:
>
> **Pre-context** ("In 2038, researchers discovered X is → Y") modifies the prompt embedding itself, potentially directing updates to new representational space, mimicking latent cause creation.
>
> **Post-context** ("X is → Y, after discoveries in 2038...") preserves the original prompt, forcing contextualized revisions onto existing representations, predicting possible interference.
>
> Our experiments test both dimensions: (a) measuring and understanding collateral damage across surprise levels (Sec.6.2->Sec6.5) and (b) evaluating whether pre-contextualization can mitigate destructive contradictions (Sec.6.1)
>
> ---
>
> In this work, we ultimately opted for varying surprise and conflict as judged qualitatively by a human. Our initial intention was indeed to link these updates to some sort of quantitative surprise or prediction error, which turned non-trivial after initial naive attempts. For example, token-level surprisal (perplexity) measures novelty but treats contradictions identically to compatible facts if both are equally unexpected tokens. We also found that gradient magnitudes were comparable whether an update extends or contradicts prior knowledge. Also given there were too many results to handle for a single paper, we decided to defer the mechanistic investigation to a follow-up work.
>
> Finally, we revised Table1 to make the distinction between update types more explicit [(you can visualize it here)](https://figshare.com/s/b633a755b65bbd873a34?file=59752601). We hope this framing better clarifies our results which suggest that "contradictions" (whether factual, code, or ethics related), rather than surprise itself, are what drive destructive collateral damage.
>
> > The datasets used are mostly LLM generated. This isn't damning, but I still always prefer to see evals which aren't circular. There could be some confound introduced by using LLMs to evaluate LLMs
>
> This is a valid concern. We partially address it with a small Wikipedia-based dataset (N=30, Appendix G.6) that doesn't rely on LLM generation. While limited in scale and using synthetic entity swaps, it shows trends consistent with our main findings (tested with gpt-4.1-nano and compared with the facts/counterfacts datasets for N=30). Developing larger, naturally-occurring counterfactual datasets is an important direction for future work.

---

> > ### Author Response · Authors · 2025-11-23
> > **Extension of results to a more conservative hyperparameters regime**
> >
> > > Questions: I recognize that a full grid search is computationally infeasible, but optimizing for one setting makes me concerned the results are general (esp. if then the results hinge on that setting, in your case the counterfact setting, being exceptional in some way). Could you try optimizing hyperparams on one other dataset, any dataset, just to sanity check that the trends stay the same and are not dependent on how you optimized?
> >
> > This is indeed reasonable. We have now added a conservative hyperparameter regime where we (1) focus on another dataset (non conflicting facts that were unknown to the model) and (2) reduce the number of training samples from 300 (which might result in overtraining) to 1, 3 or 10 training samples, using the minimum number of epochs that allows the model to learn the new knowledge. For GPT 4.1, we reduce the number of epochs from 10 to 3 (see table below).
> >
> > | Model Training | Epochs (Standard) | Epochs (Conservative) |
> > |----------------|-------------------|----------------------|
> > | Meta-Llama-3-8B LoRA | 5 | 1 |
> > | Mistral-7B LoRA | 10 | 3 |
> > | GPT-2-XL Full FT | 10 | 2 |
> > | GPT-2-XL LoRA | 20 | 7 |
> > | GPT-4.1 | 10 | 3 |
> > | GPT-4.1-mini | 10 | 3 |
> > | GPT-4.1-nano | 10 | 3 |
> >
> > We reran experiments in Table 3 using this conservative regime (more than 500 new finetunings and 400k+ LLM-judged predictions) and reported the new results in Tab. 17 in Appendix G.5. [The results for the conservative setting can be viewed here too for your convenience.](https://figshare.com/s/b633a755b65bbd873a34?file=59752604)
> >
> > Thank you again.

---

### Author Response · Authors · 2025-12-03
**Executive summary: paper overview and key contributions**

We thank the reviewers for their constructive feedback, allowing us to significantly enhance our work. In response, we have (1) tightened the link between motivation and empirical results, (2) improved presentation and reduced reliance on appendices by moving key methods and findings into the main text, (3) validated our results under more conservative training regimes, and (4) deepened our analysis with quantitative findings on transferable habits.

## Paper Overview

According to latent cause theory, biological systems use prediction error to decide between modifying existing memories and creating new ones that keep the old representations intact. LLMs lack this mechanism: gradient descent applies identical updates regardless of how surprising the new information is. In this work, we ask two questions about the consequences.

*1. What happens when LLMs encounter varying levels of surprise?*

We construct training updates across four levels: (i) known facts, (ii) semantically equivalent yet lightly surprising rephrasings, (iii) novel but compatible facts, and (iv) direct contradictions. We further create contradictions in the ethical and code domain. We find that:

- Contradictory updates (whether factual, code- or ethics-related) cause damage to totally unrelated knowledge; within-domain (e.g. fact to unrelated fact) and cross-domain.
- This damage scales monotonically with surprise level (see Table 3 below).
- Damage sometimes spills across domains *but only for contradictory updates*, e.g., training on disguised malicious code degrades factual accuracy with models producing code in response to factual queries.

*2. Can temporal linguistic framing protect against this damage?*

In biological systems, high prediction error can trigger inference of a new "latent cause" (the system concludes that the current situation is different from previous ones, and creates a separate memory trace rather than overwriting existing memory). Crucially, environmental context (a different room, a different time) establishes this separation before the surprising information arrives.

We test whether contextualization with linguistic framing can play an *analogous role*. Pre-contextualization ("In 2038, it was discovered that X --> Y") establishes temporal context before the contradictory content Y. Post-contextualization ("X --> Y, according to 2038 discoveries") attempts to contextualize after, potentially targeting the original representation.

We find that pre-contextualization prevents damage; while post-contextualization does not. This asymmetry *parallels* the biological case: just as environmental context precedes experimental attempts to overwrite the original memory, linguistic context must precede contradictory content to be protective. We hypothesize that post-contextualization fails because the model has processed the contradiction under its default representation. Together, these findings suggest that contextualization, which biological systems may rely on implicitly, is a promising direction for safe memory updating in LLMs.
These empirical findings establish a behavioral parallel with a behavioral theory (latent cause is also behavioral), calling for future mechanistic interpretability work, to understand the reasons behind the protective nature of contextualization.

---

### Table 3. Degradation scales with surprise (better presentation of old results)

| Surprise Level|Update|GPT-2 XL |Mistral |Llama|GPT-4.1 nano|GPT-4.1 mini|GPT-4.1|
|---|---|---|---|---|---|---|---|
| (i) no surprise|Initial facts|0|0|0|0|0|0|
| (ii) low surprise|Alternative|-0.18|-0.11|-0.11|-0.07|-0.10|-0.07|
| temporal| Pre-context|-0.17|-0.18|-0.33|-0.12|-0.23|-0.08|
| (iii) novel| Fictional|-0.48|-0.26|-0.51|-0.33|-0.10|-0.05|
| (iv) contradictory| Counterfacts|-0.66| -0.79 | -0.64 |-0.62|-0.80|-0.91|
| temporal|Post-context|-0.57|-0.84|-0.75|-0.69|-0.85|-0.95|

[Colored visualization](https://figshare.com/s/b633a755b65bbd873a34?file=60081239)

---

> ### Author Response · Authors · 2025-12-03
> **Summary of revisions made during rebuttal**
>
> We highlight the changes operated during the rebuttal, and related to:
> - (1) tightened the link between motivation and empirical results (asked by ZRan, padk, ECWU):
>   + Additional Section 3 with hypotheses and connection to latent cause inspiration,
>   + Highlight the connection with Table 3 showing the degradation by surprise, following the taxonomy from Section 3
> - (2) improved presentation and reduced reliance on appendices (asked by ZRan, 14mR):
>   + Adding the Table 1 in the main text, providing an [example for all 11 update types (visualization)](https://figshare.com/s/b633a755b65bbd873a34?file=59752601)
>   + Providing more details about the generation methodology of the novel dataset (Section 4)
>   + Clarify the fine-tuning methodology and empirical pipeline (Section 5)
> - (3) validated our results under more conservative training regimes (asked by 14mR):
>   + Adding a conservative regime, that needed 500+ additional fine-tunings (Appendix G.4)
>   + Additional study with Wikipedia-based validation (N=30, Appendix G.5)
> - (4) deepened our analysis with quantitative findings on transferable habits (asked by padk, ECWU):
>   + New section 6.4 in the main with a quantitative analysis (instead of qualitative in the initially submitted version)
>   + Systematic habits analysis in Appendix H with details
>
> The appendix has also been re-organized and substantially completed (e.g. third-judge to validate dataset diversity in Appendix A, generation and verification prompts for all updates, all LLM judge prompts now included, detailed partitioning of the initial knowledge of the models with a new figure in Appendix C).

---

> > ### Author Response · Authors · 2025-12-03
> > **Reviewer-by-reviewer response summary**
> >
> > ## Reviewer 14mR
> > **Status: We responded to all requests**
> >
> > > *"This was the best paper within the group of papers [...] by a decent margin, and I think it should be accepted."*
> > > *"enjoyed [...] connection to ideas from [...] neuroscience. Often I am critical of such connections if they lend themselves to over-interpretation, but [...] authors were responsible"*
> > > *"interesting points illustrating important future research directions"*
> >
> > |Request|Response| Location |
> > |---------|----------|----------|
> > | Conservative hyperparameters on different dataset | 500+ new finetunings with reduced epochs | Appendix G.4.2 |
> > | Non-LLM evaluation data | Wikipedia-based validation (N=30) | Appendix G.5|
> > | Better connection to surprise| New Sec. 3 linking surprise -->degradation| Section 3, Section 6.2|
> >
> > ---
> >
> > ## Reviewer padk
> > **Status: We responded to all concerns with new experiments and clarifications**
> >
> > > *"important and timely research direction [...] extensive fine-tuning experiments [...] results reasonably solid"*
> >
> > |Request|Response| Location|
> > |---------|----------|----------|
> > | Formalized problem definition| New Section 3 with explicit hypotheses| Sec. 3|
> > | Clearer update categorization| New Table 1 in the main text with examples per type| Tab. 1|
> > | Dataset verification details| Third judge validation (88% topic match)| App. A|
> > | Llama-3 zero accuracy anomaly| Model collapse analysis (same answer to 200 questions)| Appendix G.7|
> > | Code bleeding: benign vs harmful| Systematic analysis: **4% vs 73%**| Appendix H|
> > | Pre/post context definitions| Clarified upfront| New section 3|
> >
> > ---
> >
> > ## Reviewer ZRan
> > **Status: Responded; positive confirmation on Section 3, a missing point; addressed with new surprise table**
> >
> > > *"premise highly relevant [...] touches a deep limitation"*
> > > *"findings genuinely very interesting (surprising but sensible)"*
> >
> > **Post-rebuttal response**: *"the new chapter reads nicely and adds necessary context"*
> >
> > |Request|Response| Location|
> > |---------|----------|----------|
> > |Clear underlying hypothesis|New Section 3|Sec. 3|
> > |Link surprise levels --> results|New degradation table sorted by surprise|Sec. 6.2|
> > |Pre/post context asymmetry explanation|Prompt masking during finetuning|Sec. 3, Response|
> > |Table 4 accuracy drop clarification|Re-tested pretrained models, showed natural fluctuation|Response, Appendix G.6|
> >
> > ---
> >
> > ## Reviewer ECWU
> > **Status: Responded to all concerns**
> >
> > > *"dataset provides extensive and structured benchmark"*
> > > *"systematic definition of update types [...] supporting clear causal conclusions"*
> > > *Soundness: good*
> >
> > ### Main issues
> >
> > |Request|Response|Location|
> > |---------|----------|----------|
> > |**Novelty vs Dhingra/MuLan**|**See clarification below**|Added in Sec. 2, Response|
> > |Writing clarity|substantially reworked main and appendix|New Sec.3; Rewrote Sec. 4-6; augmented App. A, improved App. C |
> > |Habits analysis is anecdotal|Systematic LLM-judged: code bleeding, harm, length|Sec. 6.4 and new App. H|
> >
> > ### Novelty clarification (vs. Dhingra TACL'22 and MuLan NAACL'24)
> >
> > Prior work studied conservation of *same-entity* temporal facts:
> > - Train: "LeBron plays for --> Cavaliers [2010]", then "LeBron plays for --> Lakers [2018]"
> > - Test: "LeBron plays [2010]?" --> Cavaliers
> >
> > Our work tests completely unrelated entities across domains:
> > - Train: e.g. "Capital of Germany is --> London" (counterfact)
> > - Test: "Maradona is a...?" (facts), "Stealing at work is...?" (ethics), "Write a sort function" (code)
> > - Result: Catastrophic loss across **unrelated** facts, ethics, code. This loss is prevented by pre-context (but not post-context)
> >
> > Both within and cross-domain damage on *unrelated knowledge* are distinct from prior temporal knowledge work. MuLan further focused on in-context editing, while we focus on parametric memory.
> >
> > ### Other clarification requests
> >
> > |Request|Response|Location|
> > |---------|----------|----------|
> > |Add dataset examples in main|New Table 1 with examples|Tab. 1|
> > |Train/eval split clarification|writing + new figure|Section 4, Figure 3 in App. C|
> > |Topic sampling process|Detailed taxonomy (82 ethical, 61 coding topics)|App. A|
> > |Verification process|Generation + verification prompts for all types|App. A, clarified in Sec. 4|
> > |"Uncommented disguised code"|Table note in Table 1|Tab. 1, App. A|
> > |Generation details|Greedy for open models, defaults for GPT-4.1|App. D|
> > |Correctness evaluation details|LLM-as-judge with domain-specific prompts|Added to App. E, Listings 10-14|
> > |"Update dose" definition|Replaced with epochs/samples/steps|Throughout|
> > |Figure 2 axis|Clarified = retention rate (our sole metric in the paper)|Fig. 2 caption and throughout|
> > |Table 2 missing GPT|clarified GPT in Tab. 2, added GPT to Table 4|Tab. 2 and 4|
> > |Fig 1 / Tab 3 / Tab 4 relationship|Clarified|Sec. 4, Fig. 1 caption|
> > | BaselineQA / FreebaseQA unexplained|better defined the first, deferred the second to App. B.|Sec. 4 App. B|

---

### Meta-Review · Area_Chair_3oBH · 2025-12-23

**Summary:**

1. The paper is overly embellished. It seems to lack some connections between mechanisms in human memory and LLM results. The latent cause seems to lack connections with the findings of the paper. (Reviewer 14mR and ZRan)
2. Novelty: The novelty in this paper lies instead in the systematic cross-domain evaluation, which should be stated more explicitly. (Reviewer ECWU)
3. Writing and organization: A more formalized problem definition and a clearer categorization of the knowledge update types would strengthen the empirical pipeline. The writing and organization of the paper could also be improved for clarity. (Reviewer padk and ECWU)
4. Lacks some detailed analysis: The abnormal behavior of Llama-3 is not discussed in Section 5.4. Lacks a clear underlying hypothesis and deep analysis of some findings, and difficult to draw some valuable conclusions. (Reviewer padk and ZRan)
5. Missing some details about datasets: The datasets used are mostly LLM-generated. This paper lacks some details about the automated verification process that is necessary to ensure the quality and diversity of the generated dataset. Details for the data generation are missing for the train/evaluation splits. The dataset is one of the paper’s main contributions, but it is insufficiently explained in the main text.
 (Reviewer 14mR, padk, and ECWU)
6. Experiments: This paper optimizes hyperparameters by a full grid search on only one dataset, lacking generalization to other settings. Reviewer padk points out that an important comparison would be between cross-domain fine-tuning with the original data and with the counterfactual data. The breadth of the experimental setup is confusing. The distinctions between fine-tuning and evaluation splits are not clearly presented. (Reviewer 14mR, padk, ZRan, ECWU)

**Reviewer Concerns:**

The authors have diligently responded to all reviewer comments, which is appreciated. However, this paper still comes across as somewhat over-packaged, missing some necessary findings or conclusions. (Reviewer 14mR,  padk, and ZRan) Moreover, incorporating more details and refining the writing style/structural organization would enhance the overall readability and impact of the work.

**Reviewer Scores:**

Reviewer 14mR: retains 8

Reviewer padk:  retains 4

Reviewer ZRan:  retains 4

Reviewer ECWU: 2 --> 4

---

### Decision · Program_Chairs · 2026-01-26

Reject